# Graph Neural Networks For Multi-Image Matching

## Abstract

In geometric computer vision applications, multi-image feature matching gives more accurate and robust solutions compared to simple two-image matching. In this work, we formulate multi-image matching as a graph embedding problem, then use a Graph Neural Network to learn an appropriate embedding function for aligning image features. We use cycle consistency to train our network in an unsupervised fashion, since ground truth correspondence can be difficult or expensive to acquire. Geometric consistency losses are added to aid training, though unlike optimization based methods no geometric information is necessary at inference time. To the best of our knowledge, no other works have used graph neural networks for multi-image feature matching. Our experiments show that our method is competitive with other optimization based approaches.

## 1 Introduction

Feature matching is an essential part of Structure from Motion and many geometric computer vision applications. The goal in multi-image feature matching is to take 2D feature positions from three or more images and find which ones correspond to the same point in the 3D scene. Methods such as SIFT feature matching (Lowe, 2004) combined with RANSAC (Fischler & Bolles, 1981) have been the standard for decades. However RANSAC-based approaches are limited to matching pairs of images, which can lead to global inconsistencies in the matching. Other works, such as Wang et al. (2018), have shown improvement in performance by optimizing cycle consistency, i.e. enforcing the pairwise feature matches to be globally consistent.

However, these multi-view consistency algorithms struggle in distributed and noisy settings. Having image features suited for this task would help improve performance, and deep learning has revolutionized how image features are computed (Yi et al., 2016). In this paper, we want to leverage the power of deep representations in order to compute feature descriptors that are robust across multiple views.

Unfortunately, there are obstacles to applying multi-view constraints directly to deep learning. Multi-view constraints are formulated in terms of sparse features, which traditional convolutional neural nets are not designed to handle. Thus we will need a new architecture to handle such constraints. More fundamentally, deep neural networks need large amounts of labeled data to train. Consequently unsupervised training is a more practical approach. In the absence of direct supervision, the additional signal of geometric constraints can help disambiguate visually similar features and reject outliers. Thus incorporating such constraints is important in training a network to solve this task.

In this work, we address these concerns using Graph Neural Networks (GNNs). The proposed method works directly on the graph of correspondences between the image features, which is agnostic to how the correspondences were computed, thus allowing the algorithm to work in a broad class of environments. To the best of our knowledge this work is the first to apply deep learning to the multi-view feature matching problem. We use an unsupervised loss, the cycle consistency loss, to train the network. We use geometric consistency losses to aid training, though no geometric information is used at inference time. Although our network is simple, it shows promising results compared to baselines which optimize for cycle-consistency without learned embeddings, using a matrix factorization loss (Zhou et al., 2015b; Leonardos et al., 2017). Furthermore, since inference requires only a single forward pass over the neural network, our approach is faster to achieve

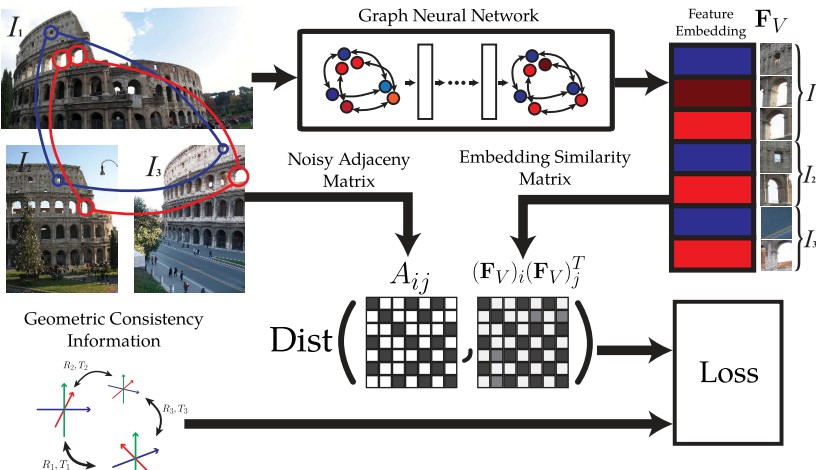

Figure 1: An illustration of the approach of this work. The Graph Neural Neural Network (GNN) (Battaglia et al., 2018) takes as input the graph of matches and then outputs a low rank embedding of the adjacency matrix of the graph. The GNN operates on an embedding over the vertices of the graph. In the figure, the GNN vertex embeddings are represented by different colors. The final embedding is used to construct a pairwise similarity matrix, which we train to be a low dimensional cycle-consistent representation of the graph adjacency matrix, thus pruning the erroneous matches. We train the network using a reconstruction loss on the similarity matrix with the noisy adjacency matrix, and thus do not need ground truth matches. In addition, we can use geometric consistency information, such as epipolar constraints, to assist training the network.

comparable accuracy than methods which must solve an optimization problem every time. We perform experiments on the Rome16K dataset (Li et al., 2010) to test the effectiveness of our method compared to optimization based methods. Our contributions in this work are:

- We use a novel architecture to address the multi-image feature matching problem using GNNs with graph embeddings.
- We introduce an unsupervised multi-view cycle consistency loss that does not require labeled correspondences to train.
- We demonstrate the effectiveness of geometric consistency losses in improving training.

## 2 RELATED WORK

### 2.1 FEATURE MATCHING

Image feature matching has a rich history of research in computer vision. Much work has be done using hand-crafted feature descriptors such as SIFT (Lowe, 2004), SURF (Bay et al., 2006), BRIEF (Calonder et al., 2012), or ORB (Mur-Artal et al., 2015). RANSAC Fischler & Bolles (1981) is the most widely used robust estimation technique to filter out outliers from the matches. The combination of RANSAC and hand-crafted feature descriptors has constituted the bulk of the matching literature for the last 40 years. More recently Suh et al. (2015) and Hu et al. (2016) have shown graph matching of the features can be added for more robust matches between images.

### 2.2 MULTI-IMAGE MATCHING

Multi-image matching has traditionally been done using optimization based methods minimizing a cycle consistency based loss (see Section 3.3). Pachauri et al. (2013) and Arrigoni et al. (2017) use the eigenvectors of the matching matrix to obtain a low dimensional embedding. However, the assumption of low Gaussian noise is not realistic. Zhou et al. (2015b) and **?** use more sophisticated optimization techniques on the matching matrix and thus produce more robust solutions. Leonardos et al. (2017) implement a distributed optimization scheme to solve for cycle consistency. Swoboda

et al. (2019) implement a convex relaxation of the low dimensional embedding problem. Shi et al. (2016) use tensor power iterations to solve the matching problem, also taking into account the intra-image matching graph. As an alternative to optimization based techniques, Tron et al. (2017) used density based clustering techniques to compute multi-image correspondence. Fathian et al. (2019) use a similar technique but formulates it as a generalized Rayleigh quotient problem to achieve better results. Moving away from feature matching, Zach et al. (2010) uses cycle-consistency-like constraints on pose graphs quite effectively. To the best of our knowledge, we are the first to use graph neural networks for multi-image matching.

### 2.3 DEEP LEARNING FOR MATCHING

Previous attempts to improve image matching techniques using machine learning have focused on learning the descriptors given ground truth correspondence from curated datasets such as Zagoruyko & Komodakis (2015); Yi et al. (2016); and Brachmann et al. (2017). Rocco et al. (2018) use a dense matching tensor with a weakly supervised loss to learn good features for matching. Zhou et al. (2019) has shown that these learned representations have limited ability for improving SfM tasks.

There are other methods to build correspondences such as Choy et al. (2016), but they only handle two-view constraints and require dense correspondences. Most similar to our work, Zhang & Lee (2019) use graph neural networks to do intra-image feature processing before doing two image inter-image feature similarity. However, their work only trains for pairs of image, while ours is explicitly trained for multi-image matching. Also similarly, Yi et al. (2018) attempts to improve correspondences by learning match probabilities for two-view RANSAC for greater robustness and speed. Like us Zhu et al. (2017) use cycle consistency in their loss; however their method is for image generation and is restricted to pairwise cycle consistency. Our method can be applied to 3 or more images. Hartmann et al. (2017) learns multi-image matching but requires heavy supervision from 3D object reconstructions, which can be difficult or expensive to obtain. Zhou et al. (2015a), unlike our method, uses dense correspondence, but uses cycle consistency to find semantic matches accross multiple views. Suwajanakorn et al. (2018) find 3 dimensional latent keypoints, trained using ground truth rotation and translation. However, their method is restricted to a limited number of object categories, which is different from the SfM setting we are considering here.

### 2.4 GRAPH NEURAL NETWORKS

Graph neural networks, true to their name, are deep neural networks operating in graph domains. Multi-image matching is novel application of them, as they are more typically used for applications such as citation networks or recommendation systems. They have received more attention recently (Bronstein et al., 2017; Defferrard et al., 2016; Kipf & Welling, 2017; Scarselli et al., 2009; Gama et al., 2018b;a; Battaglia et al., 2018). The first methods to learn neural networks over graphs were the so-called Spectral methods. They used the eigenvectors of the graph Laplacian to compute convolutions, which can learn graph specific convolution kernels (as in Bruna et al. (2013)), but require an a-priori known graph structure. Newer non-spectral methods do not require a-priori knowledge, as seen in Bronstein et al. (2017); Kipf & Welling (2017); Scarselli et al. (2009); and Gama et al. (2018a). Most of these methods use polynomials of the graph Laplacian to compute neighborhood averages. Gama et al. (2018b;a) formalize this notion and generalize it beyond the use of the graph Laplacian. To improve performance, more sophisticated aggregation techniques and global information passing can be used as discussed in Battaglia et al. (2018). The closest work on Graph Neural Networks to ours is Kipf & Welling (2016), which uses GNNs to reconstruct adjacency matrices, though not in a geometric context.

## 3 METHOD

Our goal is to learn optimal features that capture multiple image views by filtering out noisy feature matches. The input to our algorithm is a set of features and noisy correspondences, and the output is a new set of features where the pairwise similarities of these features correspond to the true matches. An outline of our approach can be seen in figure 1. We do this by training the new set of feature embeddings to be cycle consistent. We formulate this problem in terms of the correspondence graph of the features. Graphs $\mathcal{G} = (\mathcal{V}, \mathcal{E})$ have a set of vertices $\mathcal{V}$ and of directed edges $\mathcal{E} \subseteq \mathcal{V} \times \mathcal{V}$.

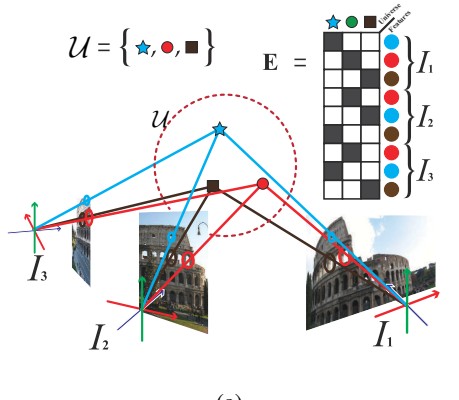
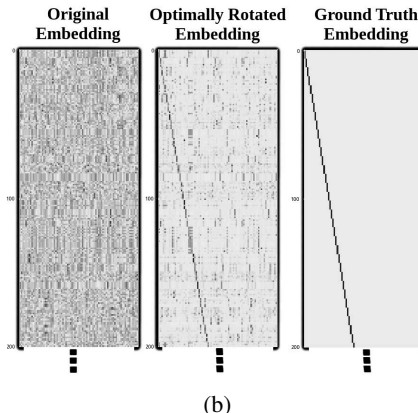

(a)                                                                    (b)

Figure 2: **(a)** An illustration of the idea of the universe of features. Each feature in each image corresponds to a 3D point in the scene. We can construct cycle consistent embeddings of the features by mapping each one to the one-hot vector of its corresponding 3D point. While there can be many features, there are fewer 3D points and thus this corresponds to a low rank factorization of the correspondence matrix. Best viewed in color. **(b)** Visualization of the learned embeddings. On the left we have the raw outputs, which are difficult to interpret. In the center, we rotated the features to best match the ground truth for a more interpretable visualization (see the end of Section 3.3). On the right, we have the ground truth embeddings, given as indicator vectors for which feature in the world the points correspond to. For the optimally rotated embedding we can see that the true embedding structure is recovered (with some noise).

For a vertex $v \in \mathcal{V}$ we use $\mathcal{N}^h(v)$ to denote the $h$-hop neighbors of $v$, with the superscript left out for 1-hop neighbors. Similarly $\mathcal{E}(v)$ is used to denote the edges associated with $v$. To denote the vertices connected to an edge $e \in \mathcal{E}$ we write $e(v_1, v_2)$.

## 3.1 CORRESPONDENCE GRAPH

We assume there is an initial set of feature matches represented as a graph $\mathcal{G} = (\mathcal{V}, \mathcal{E})$, with an associated adjacency matrix $\boldsymbol{A}$. The graph is constructed from putative correspondences of image features across images, typically constructed using feature descriptor distance (e.g. SIFT feature distance). While there are many interesting methods for computing these putative correspondences (Suh et al., 2015; Yi et al., 2018), we do not explore them in this work. Typically putative correspondences are matched probabilistically, meaning a feature in one image matches to many features in another. The ambiguity in the matches could come from repeated structures in the scene, insufficiently informative low-level feature descriptors, or just an error in the matching algorithm. Filtering out these noisy matches is our primary learning goal.

Each vertex of the graph $v \in \mathcal{V}$ is an image feature, corresponding to some ground truth 3D point $\boldsymbol{p}(v)$. Each edge $e = (v_1, v_2) \in \mathcal{E}$ is a potential correspondence. Associated with each vertex $v$ is an embedding $\boldsymbol{f}_v \in \mathbb{R}^m$, which can include the visual feature descriptor, position, scale, orientation, etc. Similarly, each edge $e$ has an associated feature $\boldsymbol{f}_e \in \mathbb{R}^p$ (in this work, initially just the weight of the feature association). We use these features as the initialization for our learning algorithm.

In the absence of noise or outliers, this graph would have a connected component for each visible point in the world, all mutually disjoint. Without noise, vertices $v$ would only match with other vertices $v'$ that correspond to the same 3D point in the scene. Since features in this case represent unique locations in the scene, no points in the same image would have edges $e$ between them. Mathematically, this can be expressed as $e = (v_1, v_2) \in \mathcal{E} \implies \boldsymbol{P}(v_1) = \boldsymbol{P}(v_2)$. In the noisy case we expect this structure to be corrupted, i.e. there are some edges $e = (v_1, v_2) \in \mathcal{E}$ such that $\boldsymbol{P}(v_1) \neq \boldsymbol{P}(v_2)$. Thus we need to prune the erroneous edges.

However, standard CNNs cannot operate on this general graph structure. Thus we cannot use standard convolutional nets to learn features for this task. Instead we use graph networks to learn feature representations on this space, which we describe in the next section.

## 3.2 Graph Neural Networks for Feature Matching

As input to our method we are given a graph $\mathcal{G} = (\mathcal{V}, \mathcal{E})$ with the features described in Section 3.1: $\boldsymbol{f}_v \; \forall v \in \mathcal{V}$ and $\boldsymbol{f}_e \; \forall e \in \mathcal{E}$. As with any neural network, GNNs have layered outputs. We describe the output of layer $k$ as $\boldsymbol{f}_v^{(k)} \in \mathbb{R}^{m_k} \; \forall v \in \mathcal{V}$ and $\boldsymbol{f}_e^{(k)} \in \mathbb{R}^{p_k} \; \forall e \in \mathcal{E}$, with the initial embeddings denoted $\boldsymbol{f}_v^{(0)} = \boldsymbol{f}_v$ and $\boldsymbol{f}_e^{(0)} = \boldsymbol{f}_e$. To aid future analysis, we will represent the features as matrices, denoting the vertex embedding matrix as $\boldsymbol{F}_V^{(k)}$ and the edge embedding matrix as $\boldsymbol{F}_E^{(k)}$. If a superscript is not specified then it refers to the final output of the network.

First we describe older methods of GNNs to give context, then we describe the method we use in this work. Many older methods assume we have the adjacency matrix $\boldsymbol{A}$ of the graph known a-priori Bruna et al. (2013), and can encode graph convolutions using the eigenvectors of $\boldsymbol{A}$ (these are known as spectral methods). However, we do not have this luxury, as the correspondence structure changes from image set to image set, and thus we use non-spectral Graph Neural Networks. Newer models use non-spectral methods, which often ultimately amount to transforming each node with learned weights then averaging each node's representation with its neighbors, known as a message pass (Kipf & Welling, 2017; Defferrard et al., 2016; Gama et al., 2018b;a). Some works such as Gama et al. use pooling operations on the vertices to make the graph smaller and thus aid computation, but as we need labels on every vertex of the original graph, we cannot use this. Most GNNs used in these works can be expressed mathematically as:

$$\tilde{\boldsymbol{f}}_v^{(k+1)} = \sigma \left( b^{(k)} + \boldsymbol{W}_0^k \boldsymbol{f}_v^{(k)} + \sum_{h=0}^{H} \sum_{v' \in \mathcal{N}^h(v)} f_{e(v,v')} \boldsymbol{W}_h^k \boldsymbol{f}_{v'}^{(k)} \right)$$

The weights/biases $\boldsymbol{W}_h^k$, $b^{(k)}$ are all learned, with no learning done on the edge weights $f_{e(v,v')}$. Note that this is just averages over $h$-hop neighborhoods, where the weights on the edges remain static through the computation. Given that we are trying to prune edges, we add features over edges to learn which ones to prune and which to keep such as in Scarselli et al. (2009).

In this work we use the method and implementation described in Battaglia et al. (2018). Battaglia et al. (2018) uses message passes between node features as well as edge features, which the model can use to prune unnecessary or erroneous edges. Therefore there is intermediate processing on the edges before information is passed to the vertices.

Mathematically, this is expressed as:

$$\tilde{\boldsymbol{f}}_{e(v_1,v_2)}^{(k+1)} = \sigma \left( a^{(k)} + \boldsymbol{U}_0^{(k)} \boldsymbol{f}_e^{(k)} + \boldsymbol{U}_1^{(k)} \boldsymbol{f}_{v_1}^{(k)} + \boldsymbol{U}_2^{(k)} \boldsymbol{f}_{v_2}^{(k)} \right) \tag{1}$$

$$\tilde{\boldsymbol{f}}_v^{(k+1)} = \sigma \left( b^{(k)} + \boldsymbol{W}_0^{(k)} \boldsymbol{f}_v^{(k)} + \sum_{e \in \mathcal{E}(v)} \boldsymbol{W}_1^{(k)} \boldsymbol{f}_e^{(k+1)} \right) \tag{2}$$

Here the learned weights are denoted $\boldsymbol{W}$ and $\boldsymbol{U}$, and the biases $a^{(k)}$ and $b^{(k)}$. In Battaglia et al. (2018), they allow for more sophisticated aggregation functions, but in this work we simply use the mean function. Each one of these steps we refer to here as a message pass, and it is analogously equivalent to an iteration in a distributed graph based optimization method. Between each of the message passes, we further process the features using MLPs.

## 3.3 Cycle Consistency

Let $\boldsymbol{M}$ be the noiseless set of matches between our features, with $\boldsymbol{M}_{ij}$ being the partial permutation representing the matches between image $i$ and image $j$. If the pairwise matches are globally consistent, then for all $i, j, k$:

$$\boldsymbol{M}_{ij} = \boldsymbol{M}_{ik} \boldsymbol{M}_{kj} \tag{3}$$

In other words, the matches between two images stay the same no matter what path is taken to get there. This constraint is known as *cycle consistency*, and has been used in a number of works to optimize for global consistency Zhou et al. (2015b); **?**; Leonardos et al. (2017). Stated in this form, there are $O(n^3)$ cycle consistency constraints to check. A more elegant way to represent cycle

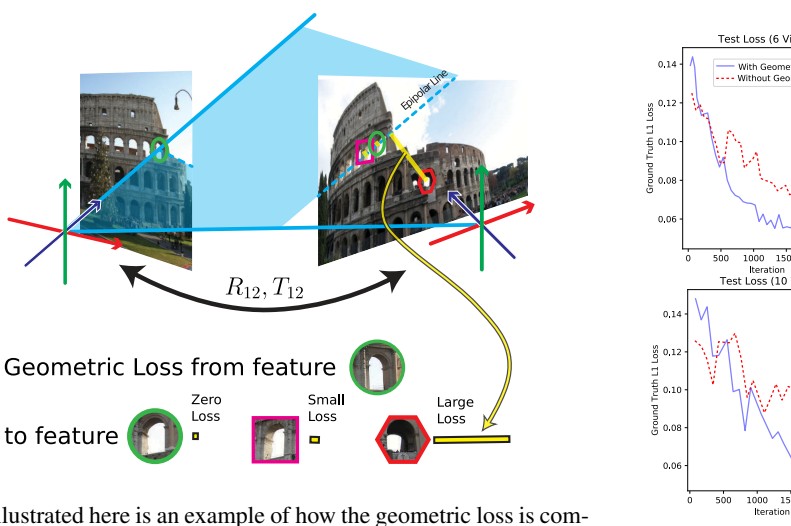

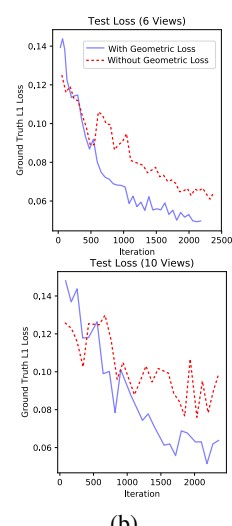

(a) Illustrated here is an example of how the geometric loss is computed for one feature.

(b)

Figure 3: **(a)** Errors are computed via absolute distance from the epipolar line, as expressed by Equation 6 via the epipolar constraint. The epipolar line is the line of projection of the feature in the first image, projected into to the second. The distance to this line on the second image indicates how likely that point is to correspond geometrically to the original feature. There can be false positives along the projected line, as shown by the square feature in the figure, but other points will be eliminated, such as the hexagonal feature. **(b)** Training curves with and without Geometric Training loss, described in 8. The geometric training loss improves testing performance. Note how training with geometric consistency losses decreases the convergence time of the network. Best viewed in color.

consistency is to first create a 'universe' of features that all images match to (see figure 2a). Then, one can match the $i^{th}$ set of features to the universe using a ground-truth matching matrix $\boldsymbol{X}_i$. Then the cycle consistency constraint becomes:

$$\boldsymbol{M}_{ij} = \boldsymbol{X}_i \boldsymbol{X}_j^\top \qquad (4)$$

This reduces the number of our constraints from $O(n^3)$ to $O(n^2)$. This was shown to be equivalent to the original definition of cycle consistency (equation 3) in Huang & Guibas (2013). We try to learn vertex embeddings $\boldsymbol{F}_V$ to approximate $\boldsymbol{X}$ - in other words the final embedding should be an encoding of the universe of features. As we do not have the ground truth matches $\boldsymbol{M}$, we approximate it using the noisy adjacency matrix $\boldsymbol{A}$ of our correspondence graph. Thus our loss would be

$$\mathcal{L}(\boldsymbol{A}, \boldsymbol{F}_V) = \mathcal{D}(\boldsymbol{A}, \boldsymbol{F}_V \boldsymbol{F}_V^\top) \qquad (5)$$

Here $\mathcal{D}$ could be an $L_2$ loss, $L_1$ loss, or many others. In this work, we use the $L_1$ loss. Note that because of this formulation, we can determine our embeddings only up to a rotation, as $\boldsymbol{F}_V R(\boldsymbol{F}_V R)^\top = \boldsymbol{F}_V R R^\top \boldsymbol{F}_V^\top = \boldsymbol{F}_V \boldsymbol{F}_V^\top$ Thus when visualizing embeddings, we rotate them to make them more interpretable (see figure 2b).

### 3.4 GEOMETRIC CONSISTENCY LOSS

In order to use geometric information, more traditional optimization based methods require the geometric information at inference time, while with learning approaches we can use it to speed up training while not needing it at inference time. Thus geometric consistency losses are one distinct advantage of our method over more traditional optimization based approaches. We use the epipolar constraint, the simplest way to add a geometric consistency loss. The epipolar constraint describes how the positions of features in different images corresponding to the same point should be related. An illustration of this is provided in figure 3a, showing how this loss can help reject erroneous points. Given a relative pose $(R_{ij}, T_{ij})$ between two cameras $i$ and $j$ (transforms $j$ to $i$) the epipolar

| Method | Same Point Similarities | Different Point Similarities |
|---|---|---|
| Ideal | $1.000 \pm 0.0000$ | $0.0000 \pm 0.0000$ |
| Initialization Baseline | $0.511 \pm 0.0168$ | $0.2560 \pm 0.2060$ |
| 5 Views, Noiseless | $1.000 \pm 0.0004$ | $0.1220 \pm 0.1670$ |
| 6 Views, Added Noise | $0.984 \pm 0.0031$ | $0.0746 \pm 0.1570$ |
| 3 Views, 5% Outliers | $0.929 \pm 0.1790$ | $0.1410 \pm 0.1480$ |
| 3 Views, 10% Outliers | $0.927 \pm 0.1790$ | $0.1400 \pm 0.1510$ |

Table 1: Results for unsupervised training on synthetic data under various noise conditions. The table plots out the weights (mean and standard deviation) of the edges reconstructed by our model, for true positive matches and true negative ones. This shows under various noise conditions that our architecture can still recover the original connectivity structure of the matching graph.

constraint on corresponding feature locations $X_i$ and $X_j$: $X_i^\top [T_{ij}]_\times R_{ij} X_j = 0$. In this work we use the two pose epipolar constraint (Tron & Daniilidis, 2014):

$$X_i^\top R_i^\top [T_j - T_i]_\times R_j X_j = 0 \tag{6}$$

The constraint assumes that the $X_k$ are calibrated (i.e. the camera intrinsics are known). Given our vertex embeddings matrix $\boldsymbol{f}_v$, we can formulate a loss between all cameras $i$ and $j$ (the vertices associated with camera $i$ denoted $\mathcal{V}(i)$):

$$\mathcal{L}_{ij,geom}(\boldsymbol{F}_V) = \sum_{v \in \mathcal{V}(i), u \in \mathcal{V}(j)} (\boldsymbol{f}_v \cdot \boldsymbol{f}_u) \left| X_v^\top R_i^\top [T_j - T_i]_\times R_j X_u \right| \tag{7}$$

For our purposes, since we use low rank embeddings $\boldsymbol{F}_V$, the loss would read (where $c(k)$ is the appropriate camera for point index $k$):

$$\mathcal{L}_{geom}(\boldsymbol{F}_V) = \text{tr}(\boldsymbol{G}^\top \boldsymbol{F}_V \boldsymbol{F}_V^\top) = \sum_{k,l} (\boldsymbol{F}_V)_k \cdot (\boldsymbol{F}_V)_l (\boldsymbol{G})_{kl} \tag{8}$$

$$(\boldsymbol{G})_{kl} = \left| X_k^\top R_{c(k)}^\top [T_{c(l)} - T_{c(k)}]_\times R_{c(l)} X_l \right|$$

## 4 EXPERIMENTS

### 4.1 SYNTHETIC GRAPH DATASET

We first test our method on synthetically generated data as a simple proof of concept. As these were simpler datasets, we the simpler edge-feature free model of (Kipf & Welling, 2017). To generate the data, we generate $p$ points, each with its own randomly generated descriptor. To create the graph, we generate random permutation matrices, with a noise applied to it after it is generated. We initialize the input descriptors using the synthetically generated ground truth descriptor, plus some added Gaussian noise. No geometric losses were added during training for these experiments. However, the method was robust in testing with different noise functions and parameters. The normalized noisy input descriptors are our baseline - they correlate with the true values but do not preserve the structure well. However, the GNN recovered the true structure very well, as shown in Table 1, showing the appropriate edge similarities. With this simple test on synthetic data passed, we now move to more challenging datasets.

### 4.2 ROME 16K GRAPH DATASET

We use the Rome16K dataset (Li et al., 2010) to test our algorithm in real world settings. Rome16K consists of 16 thousand images of various historical sites in Rome extracted from Flickr, along with the 3D structure of the sites provided by bundle adjustment. While not a standard dataset to test cycle consistency, most standard datasets have tens or hundreds images, not enough to train a GNN on. Rome16K is typically used to test bundle adjustment methods. Therefore, to use our method, we extract 6-tuples and 10-tuples of images with overlap of 80 points or more to test our algorithm, with the points established as corresponding in the given bundle adjustment output. For the initial embedding we use the original 128 dimensional SIFT descriptors, normalized to have unit $L_2$ norm,

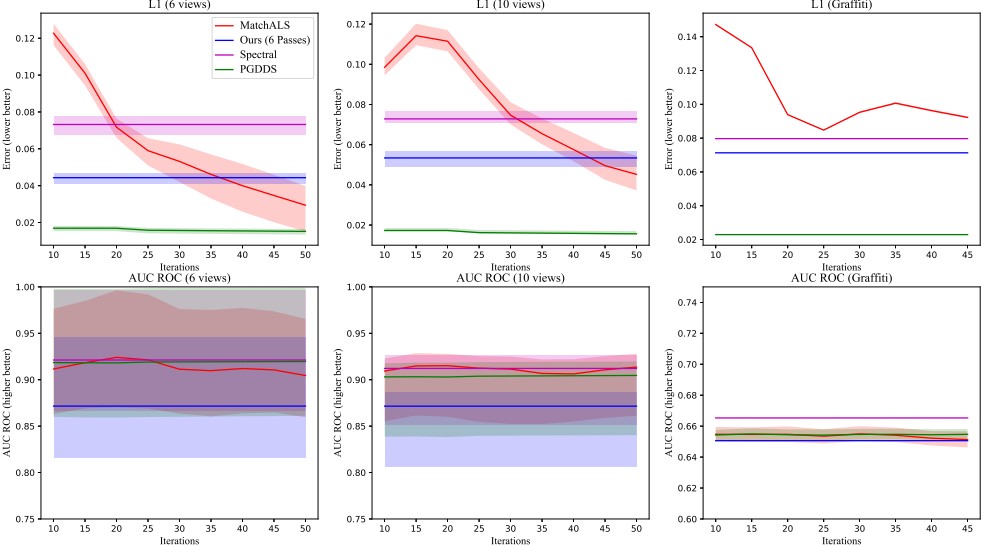

Figure 4: Plot of the losses of the baselines at different iteration numbers. The line shows the mean of the graph while the translucent coloring shows the $25^{th}$ to $75^{th}$ percentiles. The ROC AUC curves remain fairly consistent while the L1 loss goes noticibly down after more iterations. Our method compares to 35-45 iterations of MatchALS, while only having 8 message passes. PGDDS performs better than us in $L_1$ but we perform similarly in the ROC AUC metric. These results still hold even when we change domains to the Graffiti dataset (see 4.2.1).

the calibrated x-y position, the orientation, and log scale of the SIFT feature. To calibrate we use the focal lengths provided in Rome16K's data, and we assume the image center is in the center of the image (as none is provided by Rome16K). To construct the graph, we take each feature as a vertex and create edges to the 5 nearest SIFT descriptors for the other images.

For these experiments we train with the $L_1$ norm and geometric consistency losses. We evaluate on a test set using the ground truth adjacency matrix, which we compute from the bundle adjustment given by the Rome16K dataset. However, we do not train with the ground truth adjacency matrix, only with a noisy version of the adjacency matrix. We also add the geometric loss (8) which helps improve testing performance (see figure 3b). We use the $L_1$ and ROC AUC metrics to measure performance. For this method to work, we need the dimension of the embedding to be at least the number of unique points in the scene. Picking the correct number is difficult a-priori, and is a problem with all cycle consistency based methods. Here we use the ground truth dimension of the embedding to test both our method and the baselines.

The network was implemented using the code provided by Battaglia et al. (2018) using Tensorflow 1.11 (Abadi et al., 2015). Our network has 16 layers, with 8 message passing operations placed every other layer. All layers were simple Multi-layer Perceptrons, with no batch norm. The network was trained with the Adam optimizer (Kingma & Ba, 2014) with a learning rate of $10^{-4}$, with an exponentially decaying learning rate. We incorporate skip connections between the input, $6^{th}$, and $12^{th}$ layers (all possible pairs).

We compare our method to spectral and optimization based baselines with different maximum iteration cutoffs. Figure 4 illustrates this by plotting the means of various metrics and their $25^{th}$ and $75^{th}$ percentiles, with table 2 giving the exact numbers. Our network, though only using 8 message passes, has comparable accuracy to MatchALS (Zhou et al., 2015b) run 35 to 45 iterations, with an equivalent message passing step at each phase. Although our method does not outperform the Projected Gradient Descent - Doubly Stochastic (PGDDS) (Leonardos et al., 2017) method, we perform comparably to them in the ROC AUC metric.

| Method (6 Views) | $L_1$ | $L_2$ | Area under ROC | Time (sec) |
|---|---|---|---|---|
| MatchALS 15 Iterations | $0.101 \pm 0.008$ | $0.022 \pm 0.004$ | $0.918 \pm 0.073$ | $0.074 \pm 0.008$ |
| MatchALS 35 Iterations | $0.046 \pm 0.016$ | $0.010 \pm 0.005$ | $0.910 \pm 0.072$ | $0.139 \pm 0.041$ |
| MatchALS 50 Iterations | $0.029 \pm 0.017$ | $0.008 \pm 0.005$ | $0.905 \pm 0.068$ | $0.260 \pm 0.048$ |
| PGDDS0 15 Iterations | $0.017 \pm 0.002$ | $0.007 \pm 0.001$ | $0.918 \pm 0.087$ | $0.796 \pm 0.147$ |
| PGDDS0 25 Iterations | $0.016 \pm 0.002$ | $0.007 \pm 0.002$ | $0.919 \pm 0.087$ | $1.670 \pm 0.328$ |
| PGDDS0 50 Iterations | $0.015 \pm 0.002$ | $0.006 \pm 0.002$ | $0.920 \pm 0.087$ | $3.363 \pm 0.528$ |
| Spectral | $0.073 \pm 0.006$ | $0.027 \pm 0.003$ | $0.921 \pm 0.083$ | $0.036 \pm 0.005$ |
| **GNN (ours)** | $0.044 \pm 0.005$ | $0.031 \pm 0.005$ | $0.872 \pm 0.081$ | $0.765 \pm 0.046$ |
| Method (10 Views) | $L_1$ | $L_2$ | Area under ROC | Time (sec) |
| MatchALS 15 Iterations | $0.114 \pm 0.008$ | $0.028 \pm 0.004$ | $0.915 \pm 0.051$ | $0.142 \pm 0.009$ |
| MatchALS 35 Iterations | $0.065 \pm 0.009$ | $0.013 \pm 0.003$ | $0.907 \pm 0.053$ | $0.355 \pm 0.073$ |
| MatchALS 50 Iterations | $0.045 \pm 0.012$ | $0.011 \pm 0.004$ | $0.914 \pm 0.051$ | $0.455 \pm 0.022$ |
| PGDDS 15 Iterations | $0.017 \pm 0.001$ | $0.008 \pm 0.001$ | $0.903 \pm 0.061$ | $1.225 \pm 0.159$ |
| PGDDS 25 Iterations | $0.016 \pm 0.001$ | $0.007 \pm 0.001$ | $0.904 \pm 0.061$ | $2.637 \pm 0.357$ |
| PGDDS 50 Iterations | $0.016 \pm 0.001$ | $0.007 \pm 0.001$ | $0.905 \pm 0.061$ | $6.116 \pm 1.009$ |
| Spectral | $0.073 \pm 0.005$ | $0.029 \pm 0.002$ | $0.912 \pm 0.057$ | $0.081 \pm 0.021$ |
| **GNN (ours)** | $0.053 \pm 0.006$ | $0.035 \pm 0.005$ | $0.872 \pm 0.061$ | $2.438 \pm 0.070$ |

Table 2: Results on Rome16K Correspondence graphs, showing the mean and standard deviation of the $L_1$ and $L_2$. Our method was not trained on ground truth correspondences but using unsupervised methods and geometric side losses. Thus we test against ground truth correspondence graph adjacency matrices computed from the bundle adjustment output. Our method performs better than 35 iteration of the MatchALS (Zhou et al., 2015b) method, but does not perform as well as 50 iterations. We perform better than a simple eigenvalue based method (Pachauri et al., 2013). Note that we perform much better in $L_1$ performance rather than $L_2$, as we optimized the network weights using an $L_1$ loss.

### 4.2.1 GRAFFITI DATASET

We run our trained model on the more Graffiti Dataset from the Affine Covariant Regions dataset (formatted to be able to be input to our model properly). The Graffiti Dataset is the most common benchmark used in feature matching algorithms (e.g. Leonardos et al. (2017); Zhou et al. (2015b)). The results are shown in figure 4 in the rightmost figure. As the graffiti dataset is very small (only 6 views total), we were not able to train on it. We randomly permute the intra-image order of the features to add some variance - by design the GNN outputs the same result each time, while the optimization methods have a very small amount of variance. The transferred results of Graffiti are similar to the test error of Rome16K - smaller $L_1$ error and comparable ROC error. This shows that the GNNs trained in Rome16K generalize similarly to the optimization based methods.

## 5 CONCLUSION

We have shown a novel method for training feature matching using GNNs, using an unsupervised cycle consistency loss and geometric consistency losses. We have demonstrated end-to-end trainable GNNs have comparable performance the traditional optimization-based baselines. For future work, we will investigate robust losses for better outlier rejection, and using higher order geometric constraints, such as the tri-focal tensor, as additional loss terms. With this new architecture, we have the capability of training multi-image matching pipelines end to end, thus allowing us to train for image features explicitly for this task. We can extend this to distributed settings where we can train for matching images from multiple distributed agents.

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

## A    More Detail on GNN Architecture

All experiments were run with a 12 layer GNN with the ReLU nonlinearity and skip connections. The feature vector lengths were 32, 64, 128, 256, 512, 512, 512, 512, 512, 512, 1024, 1024, with skip connections between layers 1 and 6, 6 and 12, and 1 and 12. All were trained with the Adam optimizer (Kingma & Ba, 2014) and a learning rate $10^{-4}$ The network was implemented in Tensorflow (Abadi et al., 2015), version 1.11.

## B    Computing Geometric Loss

To compute the Geometric Loss during training we use the given rotations and translations from Rome16K to compute the relative poses. In practice, one could just use the relative poses computed. We build up an intermediate Essential Matrix representations $P, Q \in \mathbb{R}^{3pv \times 3}$, with $v$ being the number of views and $p$ being the number of points. The matrix $P$ is defined by the block matrix representation $P_i = [T_{c(i)}]_\times R_{c(i)} X_i$. The matrix $Q$ is defined by the block matrix representation $Q_i = R_{c(i)} X_i$. We build the Essential matrices using $E = Q^\top P + P^\top Q$, which has 3 by 3 blocks

$$
\begin{aligned}
E_{ij} &= (R_{c(i)} X_i)^\top [T_{c(j)}]_\times R_{c(j)} X_j + ([T_{c(i)}]_\times R_{c(i)} X_i)^\top R_{c(j)} X_j \\
&= X_i^\top R_{c(i)}^\top [T_{c(j)}]_\times R_{c(j)} X_j - X_i^\top R_{c(i)}^\top [T_{c(i)}]_\times R_{c(j)} X_j \\
&= X_i^\top R_{c(i)}^\top [T_{c(j)} - T_{c(i)}]_\times R_{c(j)} X_j
\end{aligned}
$$

Note that this is the same as equation 6 and thus we have our relative pariwise constraints.

