# OpenReview forum: "Graph Neural Networks For Multi-Image Matching"
_ICLR.cc/2020/Conference — Reject_

### Official Review · AnonReviewer2 · 2019-10-21
**Official Blind Review #2**

**Rating:** 6

**Review:**

Authors provide a novel approach for outlier detection of SIFT feature matchings. They construct a graph by connecting each SIFT feature to its 5 nearest neighbors initially. Then optimize a regression loss to find the matching between the 2d keypoints and 3d universal points. Hence applying cycle-consistency to figure out image matchings.

Their formulation of the problem as a GNN pruning is brilliant and widens the path for future research in the feature matching field. They also incorporate epipolar line constraints as a regularizer for their training. Experiments show effectiveness of adding the epipolar constraints.

Their experiments show that this is a promising approach, but probably requires further research to achieve state of the art results.

I believe this work is a valuable and novel method for pruning the sift feature matches. It is in an early but acceptable stage. Adding extra regularizers on F_v (to make it one-hot?) would be a promising first step. Also it has been shown that GNNs performance deteriorates with increased depth. There are recent developments in GNNs that alleviate the oversmoothing problem. Maybe switching to these architectures would enable this work to try 15 pass GNNs.

Question: How do you tune the hyper-parameters? (learning rate, number of layers, etc)

Improvement: In the experiment section explain the specifics of the geometric loss, how camera calibration, etc is calculated.

**Experience Assessment:**

I have read many papers in this area.

**Review Assessment: Checking Correctness Of Derivations And Theory:**

I carefully checked the derivations and theory.

**Review Assessment: Checking Correctness Of Experiments:**

I carefully checked the experiments.

**Review Assessment: Thoroughness In Paper Reading:**

I read the paper thoroughly.

---

> ### Author Response · Authors · 2019-11-14
> **Response to Official Blind Review #2**
>
> - “Adding extra regularizers on F_v (to make it one-hot?) would be a promising first step.”
>
> Indeed this would mean to enforce the image to universe mapping to be a partial permutation. We could even add a differential step that rotates the “soft” image to universe mapping to a partial permutation. We preferred to avoid any of those steps in order to keep the soft mapping as a feature representation which we can prune in order to compute hard matches. But we agree that we have to investigate how such a loss term (or the rotation step) would affect the final matching outcome and we will definitely add it to our experiments.
>
> - “Also it has been shown that GNNs performance deteriorates with increased depth. There are recent developments in GNNs that alleviate the oversmoothing problem. Maybe switching to these architectures would enable this work to try 15 pass GNNs.”
>
> This is a good observation, as we indeed noticed that after 12 passes of the GNN their were diminishing returns even with skip connections. Looking over such architectures will be important for future work.
>
> - “Question: How do you tune the hyper-parameters? (learning rate, number of layers, etc)”
>
> For the architecture, we searched over increasing sized networks, starting with 4 layers and going up to 15, searching various numbers of hidden nodes. We also explored various extensions such as Graph Convolutional Networks, Graph Attention Networks, and (what we finally decided on) Graph Networks from Deepmind’s graph network library. For other hyperparameters, we searched in log-linear space.
>
> - “Improvement: In the experiment section explain the specifics of the geometric loss, how camera calibration, etc is calculated.”
>
> We will add details on the calibration in the paper, and how the geometric loss is computed into the appendix.

---

### Official Review · AnonReviewer3 · 2019-10-22
**Official Blind Review #3**

**Rating:** 3

**Review:**

The paper presents graph neural network approach for learning multi-view feature similarity.
The input data is local feature descriptors (SIFT), keypoint location, orientation and scale. The objective is to learn embedding such that features corresponding to the same 3d location will have similar embeddings, while different - far away. Such embedding distance matrix is called "feature universe" in the paper.
Instead of ground truth correspondence matrix, authors use smth called "noisy adjacency matrix", although it was not clear to me, what does it mean precisely. Loss is also augmented with epipolar distance reprojection error.
Training and testing is performed on Rome16k dataset.

Overall I find paper hard to follow and weak on experiment side. Comments and questions:

 1) There is no proper (or any) traditional baseline, which is: match SIFT features, apply cross-consistency and SNN ratio thresholds, run RANSAC, throw away all the inconsistent things. See COLMAP, Bundler or any other Visual SfM/MVS pipeline. Moreover, Hartman et.al. method is cited, but not compared to, because it needs "3d reconstruction" for supervision. Here is my second objection.

 2) Paper states that the method is unsupervised. Yet, it is based on known scene geometry (R and t) , which is typically obtained via 3d reconstruction pipeline. Once you do it,  you actually have ground truth correspondences, a lot of them. I don`t understand, why not use them.

 3) Experimental validation is also weird. Method was trained on Rome16k and tested on it as well. No other datasets were used besides Graffity sequence (see below). The metric is L1 norm on adjecency matrix instead of some.

 4) Regarding "feature universe". It is clear, that one cannot fit (or can?) the full n_features x n_3d_points matrix into GPU memory, as it should be super huge matrix. No details were given on how such problems are tackled.

 5) The paper, unfortunately, has a number of side claims, which are false and actually irrelevant to the paper core. Just to list a few: "deep learning has revolutionized how image features are computed (Yi et al., 2016)." It has not. SIFT is still quite the gold standard:  (To Learn or Not to Learn: Visual Localization from Essential Matrices, https://arxiv.org/abs/1908.01293, https://image-matching-workshop.github.io), besides that cited LIFT method was not used in practice because of being super slow.
 - "More fundamentally, deep neural networks need large amounts of labeled data to train. In the case of multi-image feature matching, one would need hand-labeled point correspondences between images, which can be difficult and expensive to obtain.".
  Nobody hand-labels correspondences. Instead, one uses runs 3d reconstruction pipeline with densification like COLMAP to obtain dense depth map, what where one can get multiview correspondences (e.g., MegaDepth: Learning Single-View Depth Prediction from Internet Photos https://arxiv.org/abs/1804.00607)
  - "Typically putative correspondences are matched probabilistically, meaning a feature in one image matches to many features
in another. The ambiguity in the matches could come from repeated structures in the scene, insuffi-
ciently informative low-level feature descriptors, or just an error in the matching algorithm. Filtering
out these noisy matches is our primary learning goal.". Typically, one-to-many matches are just thrown away (e.g from Bundler SfM paper "Modeling the World from Internet Photo Collections", Sec.4.2: "If a track contains more than one keypoint in the same image, it is deemed inconsistent.
We keep consistent tracks containing at least two keypoints for the next phase of the reconstruction procedure").

 6) Citations are sometimes weird. E.g. part of OxfordAffine dataset (http://www.robots.ox.ac.uk/~vgg/research/affine/)  is referred as Graffity without any reference at all to the dataset itself (???), but with references to two irrelevant works which are testing on it. Why benchmark sycle consistency of such a small dataset of a flat surfaces? Then one could use Fountain sequence, at least, which has some non-planar structures on it.


 Minor Comments:

  - Table 1 is hardly readable because of scientific notation used.


****
After rebuttal update.

I am increasing my score a bit, but still think that paper is not is the shape for publishing. Method itself might be good, but evaluation is still bad, and what is worse - authors haven`t even tried to improve it.

>At the end of the review period it will be revealed that we have 20 years of publication history in structure from motion and visual odometry.

This, unfortunately, does not help the current paper.

>We do in fact test on data not in our training set: we never trained on Graffiti.
Graffity is still only 6 (six!!!) images, which are related by homography. One could show results, at least for HSequences (118 * 6) images or add other dataset. Since, it is not training, it could be done quite fast.

> However, the focus of our paper is a novel feature representation, the formulation using a GNN framework, and the self-supervised losses. Overall, our paper is a novel approach to learning feature representations, a topic of great interest to the ICLR community, rather than a new structure from motion system that has to prove its superior performance over the current state of the art.

I completely agree with it and would be happy to accept such paper to any kind of workshop.

**Experience Assessment:**

I have published one or two papers in this area.

**Review Assessment: Checking Correctness Of Derivations And Theory:**

I did not assess the derivations or theory.

**Review Assessment: Checking Correctness Of Experiments:**

I carefully checked the experiments.

**Review Assessment: Thoroughness In Paper Reading:**

I read the paper thoroughly.

---

> ### Author Response · Authors · 2019-11-14
> **Response to Official Blind Review #3**
>
> We thank R3 for the elaborate comments and we will try to resolve any questions and respond to criticism.
>
> 0. “Instead of ground truth correspondence matrix, authors use smth called "noisy adjacency matrix", although it was not clear to me, what does it mean precisely. Loss is also augmented with epipolar distance reprojection error.“
>
> The inter-image correspondence matrix, which we call the noisy adjacency matrix, is the adjacency matrix of the bi-partite graph of two view correspondences and it is called noisy because  we keep the top-5 nearest SIFT features as potential correspondences. These matrices and the epipolar geometries are the only input during training. The correspondence matrices are the only input during testing.
>
> 1. Indeed, we did not compare with a baseline SfM pipeline because the purpose was to evaluate multi-view correspondences. Several other papers on multi-view matching (Pachauri et al., Zhu et al., Hu et al.) do not use SfM baselines for evaluation but only ground truth correspondences. We could have cleaned two-view matches with RANSAC but we wanted to show that we can learn representations that are resilient to outliers.
>
> 2. Knowing inter-frame R,T does NOT give you ground-truth correspondences. It only constraints correspondences along epipolar lines. This is different than using transformations from world to camera which we do NOT use. Using R and T during training is a first step used by other keypoint learning papers, among  the most prominent see KeypointNet in Suwajanakorn, Snavely et al. NeurIPS 2018 (we will cite it).
>
> 3. We have isolated parts of Rome16K for testing and Rome has definitely a higher variability in appearances than for example KITTI. Graffiti is the standard baseline test  for two-view and multi-view matching.
>
> 4. This is true for all multi-view methods and that’s why we use a sliding window with a varying set of images.
>
> 5. We will retract our indeed not necessary claims. We fully agree with the reviewer  that SfM has not benefited yet from learnt representations and we will cite the learn or not to learn paper (but note that the feature learning experiments in that paper refer only to image pairs). However, the main bottleneck in any SfM problem is the correspondence problem and we should continue doing research on representations and correspondence learning, last but not least ICLR is for this purpose, not to advance SfM. We will also eliminate the “hand-labeled” sentence, we agree with the reviewer. We disagree with the reviewer that one-to-many correspondences should be thrown away. This is the current practitioners’ state of the art and should be so but from the perception point of view we should be able to build a system that can disambiguate them through multiple view consistency.
>
> 6. We just used Graffiti because it is used in the other cycle consistency papers so that we can compare results. We will correct the citations.
>
> 7. We will improve the readability of all tables.

---

> > ### Comment · AnonReviewer3 · 2019-11-14
> > **Quick reply**
> >
> > >We could have cleaned two-view matches with RANSAC but we wanted to show that we can learn representations that are resilient to outliers.
> > That is cool, but you need to demonstrate that your framework >= 2view  ransac
> >
> > >Re: Knowing inter-frame R,T does NOT give you ground-truth correspondences.
> > Running COLMAP with "dense" flag DOES give you GT corrrespondences.  Check MegaDepth paper. Besides it, it is literally, what 3d reconstruction for: gives one 3d points (related to 2d points in images)
> >
> > >This is different than using transformations from world to camera which we do NOT use
> > Why? I mean, to get R,t you need to run COLMAP anyway. Why do you don`t use the full data?
> >
> > >We have isolated parts of Rome16K for testing and Rome has definitely a higher variability in appearances than for example KITTI.
> >
> > Why not train on Rome and test on smth else from PhotoTourism datasets?
> >
> > >We disagree with the reviewer that one-to-many correspondences should be thrown away. This is the current practitioners’ state of the art and should be so but from the perception point of view we should be able to build a system that can disambiguate them through multiple view consistency.
> >
> > Thus you need to show benefit from not throwing them away, but using in your framework. I don`t see such benefit now.

---

> > > ### Author Response · Authors · 2019-11-15
> > > **Follow up to Official Blind Review #3**
> > >
> > > Dear R3,
> > >
> > > RANSAC and more generally non-learning based SfM methods are very mature and highly tuned methods.  Here, our goal is to demonstrate the potential of unsupervised learning representations in multi-image matching rather than achieve state-of-the-art performance.  It should be pointed out that in other computer vision problems learning methods initially underperformed those of their mature non-learning incumbents but were eventually surpassed, e.g., optical flow and depth estimation.  While utilizing GT correspondences can be helpful, we do not use them because having pairwise transformations is a more general assumption regarding datasets than having full 3D points and world-to-camera transformations.  For instance, R, T could be recovered from other sensing modalities (e.g., Lidar, radar, etc.) or from stereo calibration.  We use less measurements during training and empirically demonstrate this as an advantage of our approach.  Indeed, evaluating on the PhotoTourism dataset as well as comparing with classic pairwise outlier rejection are logical next steps. However, the focus of our paper is a novel feature representation, the formulation using a GNN framework, and the self-supervised losses. Overall, our paper is a novel approach to learning feature representations, a topic of great interest to the ICLR community, rather than a new structure from motion system that has to prove its superior performance over the current state of the art.

---

> > > > ### Comment · AnonReviewer3 · 2019-11-15
> > > > **Quick reply2**
> > > >
> > > > Dear authors,
> > > >
> > > > I agree with you and retract my comments regarding the sota requirement. Indeed, for the first attempts it should be necessary. Also I apologize for my harsh tone.
> > > >
> > > > Yet I would like to defend my point about evaluation as a domain expert.
> > > > I observe a similar problem across 3d-rec related domains and learning methods. Which is:
> > > >
> > > > 1) someone from ML proposes nice idea (like you), but
> > > > 2) (probably) because of lack of domain knowledge, evaluate it in a biased way.
> > > > 3) Other researchers from ML take this evaluation protocol as granted and continue to (unconsciously maybe) mislead the public.
> > > >
> > > > Examples:
> > > >
> > > > (a)PointNet-like approaches report accuracy, not mIOU for pcl segmentation. Accuracy is much less strict metric and not much correlate to the practical performace.
> > > > (b) Local feature evaluation started with https://lmb.informatik.uni-freiburg.de/Publications/2014/FDB14/1405.5769v1.pdf
> > > > Used metric is misleading and actual performance is other way round
> > > > (c) Evaluation of learned analogues to RANSAC, where RANSAC comes untuned and bad implemented.
> > > >
> > > > The problem with your response is that I have got an impression, that you are not going to actually provide proper baselines or evaluate not on a dataset, you are trained on. Follow-up papers will say the same, as you said me:
> > > >
> > > > " We just used Graffiti because it is used in the other cycle consistency papers so that we can compare results. "
> > > >
> > > >  If I am wrong - you still have time to correct me.
> > > >
> > > > Best regards, R3.

---

> > > > > ### Author Response · Authors · 2019-11-15
> > > > > **Second Follow up to Official Blind Review #3**
> > > > >
> > > > > At the end of the review period it will be revealed that we have 20 years of publication history in structure from motion and visual odometry.  As domain experts, we resonate with the reviewer’s feelings about evaluation metrics in geometric problems but we are clear and transparent that we are targeting only the correspondence problem from multiple views. We used among others the standard ROC metric introduced by Mikolajczyk and Schmid (PAMI, 2004). We do in fact test on data not in our training set: we never trained on Graffiti. Our plan for future work is to investigate including the epipolar estimation in the pipeline making the approach fully self-supervised, compare with standard outlier rejection baselines, and extend testing on Phototourism datasets.

---

### Official Review · AnonReviewer1 · 2019-10-26
**Official Blind Review #1**

**Rating:** 3

**Review:**

This paper proposes a multi-image matching method using a GNN with cyclic and geometric losses. The authors use vertex embeddings to exploit cycle consistency constraints in an efficient manner, then the GNN is used to learn the appropriate embeddings in an unsupervised fashion. The geometric consistency loss is used to aid training in addition. Experimental evaluation shows its performance compared to MatchALS (Zhou et al., 2015) and PGDDS (Leonardos et al., 2016).

I think this paper has some potential but has not matured yet. My main concerns are as follows.

1) Cyclic consistency terms
The way of learning vertex embeddings using cycle consistency in an unsupervised manner is interesting, but I'm not sure whether we should call it real cycle consistency. The authors assume a cycle going from a vertex to its embeddings, coming back to the vertex. Given two vertices, it has the form of Eq. (4), which is called cycle consistency constraints in this paper. But, it's in effect nothing but the dot product similarity between two vertex embeddings. I agree that this can be interpreted as a type of cycle constraint, but does not involve any real cycle at the end. This needs to be justified.

2) Unsupervised learning
The proposed, so-called, cycle constraint terms are learned using noisy adjacency matrix (as pseudo labels) from initial matches. This would be sensitive to the quality of the inial matches, so needs to be analyzed by experiments, which are not done. And, the effect of geometric consistency term is not clear at all in the experiments. Does it help? then how much? Some ablation studies are required.

3) Experimental comparison
While there exist many related papers on multi-image matching, the authors compared only to two methods, and the performance gain is not significant. The overall results are not convincing. See more related papers in the following.

Zhou et al., FlowWeb: Joint Image Set Alignment by Weaving Consistent, Pixel-Wise Correspondences, CVPR15
Swoboda et al, A convex relaxation for multi-graph matching, CVPR2019
Shi et al., Tensor Power Iteration for Multi-Graph Matching, CVPR2016
Yan et al., Multi-Graph Matching via Affinity Optimization with Graduated Consistency Regularization, TPAMI16


**Experience Assessment:**

I have published in this field for several years.

**Review Assessment: Checking Correctness Of Derivations And Theory:**

I assessed the sensibility of the derivations and theory.

**Review Assessment: Checking Correctness Of Experiments:**

I carefully checked the experiments.

**Review Assessment: Thoroughness In Paper Reading:**

I read the paper at least twice and used my best judgement in assessing the paper.

---

> ### Author Response · Authors · 2019-11-14
> **Response to Official Blind Review #1**
>
> We would like to address your three main concerns:
>
> (1) The low-rank constraint (eq. 4) was first introduced by Q.-X. Huang and L. Guibas (Consistent shape maps via semideﬁnite programming, 2013) and was shown to be equivalent to the “literal” cycle consistency of (eq. 3). There is a parallel to geometric transformations: Eq. 3 expresses the composition of relative poses (like local coordinate systems) which is equivalent to estimating the global pose of every coordinate system to an absolute coordinate system (universe). A row in X represents a soft assignment of universe features for an image feature in image i (or a matching probability vector), so the inner product between a row x_i and a row x_j is a soft matching value. It indeed involves a real cycle because of the sufficiency condition of (3) for (4) shown in Huang and Guibas (Consistent Shape Maps via Semideﬁnite Programming).
>
> (2) We actually do not use any hard assignments for initial matches. We use the K best matches and use the cos-distance of SIFT features. So this includes many one-to-many ambiguous matches and we could definitely vary the K to see its effect on the final representation (we have a variation of noise in synthetic data experiments in Table 1).
> Indeed, and we are honest about this in the conclusion that our results are comparable to optimization methods. However, we believe that ICLR rewards novelty of representations and that the proposed work is novel and different than traditional image feature embeddings.
>
> (3) We will cite the four references. Swoboda’s convex approach,  Shi’s Tensor Power Iteration, and Yan’s Affinity optimization are all directly relevant to the baselines we compared and we will definitely add them to the literature as well as the Flowweb which establishes dense correspondences and keypoints are a side-product of pooling.

---

### Author Response · Authors · 2019-11-14
**General Response to Reviewers**

We thank all reviewers for the helpful feedback and we provide here a high level summary of our response. With respect to Reviewer 1 (WR), we clarify two misunderstandings on cycle consistency and unsupervised learning. We admit that we could have compared with more approaches on the experiments. With respect to Reviewer 3 (R), we would like to emphasize that we proposed a novel feature representation for capturing cycle consistency across multiple views and geometric constraints. Our goal was not to provide a competitive representation for SfM/Bundle Adjustment. Reviewer 2 (WA) appreciates the novelty of the representation. The global sentiment is that the work is at early stages experimentally. We agree but we believe that ICLR is a stage for proposing novel learnt representations rather than competitive results on vision tasks like SfM. The beauty of representation research is that you cannot predict where exactly they will be useful. We believe that our framework uses the right language (graphs) for feature matching and by incorporating the reviewers’ comments we will have also a much clearer presentation of the meaning of the embeddings and the corresponding losses. We have updated the paper with the changes we made highlighted in red.

---

### Decision · Program_Chairs · 2019-12-19

**Decision:**

Reject

**Comment:**

The paper proposes a method for learning multi-image matching using graph neural networks. The model is learned by making use of cycle consistency constraints and geometric consistency, and it achieves a performance that is comparable to the state of the art. While the reviewers view the proposed method interesting in general, they raised issues regarding the evaluation, which is limited in terms of both the chosen datasets and prior methods. After rounds of discussion, the reviewers reached a consensus that the submission is not mature enough to be accepted for this venue at this time. Therefore, I recommend rejecting this submission.